# Genetic Background and Molecular Mechanisms of Juvenile Idiopathic Arthritis

**DOI:** 10.3390/ijms24031846

**Published:** 2023-01-17

**Authors:** Saverio La Bella, Marta Rinaldi, Armando Di Ludovico, Giulia Di Donato, Giulio Di Donato, Vincenzo Salpietro, Francesco Chiarelli, Luciana Breda

**Affiliations:** 1Paediatric Department, University of Chieti “G. D’Annunzio”, 66100 Chieti, Italy; 2Paediatric Department, Buckinghamshire Healthcare NHS Trust, Aylesbury-Thames Valley Deanery, Aylesbury HP21 8AL, UK; 3Paediatric Department, University of L’Aquila, 67100 L’Aquila, Italy

**Keywords:** juvenile idiopathic arthritis, JIA, molecular mechanisms, paediatric rheumatology

## Abstract

Juvenile idiopathic arthritis (JIA) is the most common chronic rheumatic disease in the paediatric population. JIA comprises a heterogeneous group of disorders with different onset patterns and clinical presentations with the only element in common being chronic joint inflammation. This review sought to evaluate the most relevant and up-to-date evidence on current knowledge regarding the pathogenesis of JIA subtypes to provide a better understanding of these disorders. Despite significant improvements over the past decade, the aetiology and molecular mechanisms of JIA remain unclear. It has been suggested that the immunopathogenesis is characterised by complex interactions between genetic background and environmental factors that may differ between JIA subtypes. Human leukocyte antigen (HLA) haplotypes and non-HLA genes play a crucial role in the abnormal activation of both innate and adaptive immune cells that cooperate in causing the inflammatory process. This results in the involvement of proinflammatory cytokines, including tumour necrosis factor (TNF)α, interleukin (IL)-1, IL-6, IL-10, IL-17, IL-21, IL-23, and others. These mediators, interacting with the surrounding tissue, cause cartilage stress and bone damage, including irreversible erosions. The purpose of this review is to provide a comprehensive overview of the genetic background and molecular mechanisms of JIA.

## 1. Introduction

Juvenile idiopathic arthritis (JIA) is the most common chronic rheumatic disease in childhood [1]. Under the generic umbrella term “JIA” are currently encompassed all forms of chronic arthritis with unknown aetiology, starting before the age of 16 years. Epidemiological studies estimate an incidence rate of about 1.6 to 23/100,000/year and a prevalence of about 16–150/100,000 in European countries, with a higher incidence rate in females (10/100,000) compared to males (5.7/100,000) [1,2]. JIA represents a heterogeneous family of distinct paediatric rheumatological disorders caused by a variety of genetic and pathophysiological mechanisms (Table 1) [3]. The International League of Associations for Rheumatology (ILAR) has defined seven different subtypes of JIA [4]. Noteworthily, oligoarticular JIA (oJIA) is the most common in European countries, whereas polyarticular JIA (pJIA) is most often seen in India, Australia, New Zealand, Costa Rica, and South Africa, enthesitis-related arthritis (ErA) in Asia, and, finally, systemic JIA (sJIA) in Japan [1,5,6]. 

Growing evidence suggests that while some of these JIA subtypes are homogeneous and have a counterpart in the adult population, others are heterogeneous and occur exclusively in children, and thus they do not represent the childhood onset of an adult illness [1,3,8]. Although onset, clinical presentation, and disease course differ in JIA subtypes, the development of a chronic inflammatory process within synovial joints is a common characteristic [1]. New classification criteria are in progress to provide a better evidence-based categorisation of the chronic arthritis belonging to the broad JIA family [3]. The pathogenesis of JIA is characterised by a disrupted balance between regulatory cells, mostly represented by T regulatory (T reg) cells, and effector cells, that include CD4+ T helper (Th) cells, such as Th1 and Th17. This disequilibrium is caused by complex interactions between genetic and environmental factors, and nowadays the literature is showing a growing interest in this field (Figure 1) [9]. Joints involved are characterised by the infiltration of multiple different immunocompetent cells that secrete inflammatory mediators. In the synovial fluid, active neutrophils with an altered functionality in combination with defective synovial monocytes and macrophages that show an impaired phagocytosis ability have been observed, confirming the key role of the innate immune system in the pathogenesis of JIA [10,11]. Furthermore, the involvement of proinflammatory cytokines has been recorded both in blood and synovial fluid, where high levels of tumour necrosis factor (TNF)α, interleukin (IL)-1, IL-6, IL-10, IL-17, IL-21, IL-23, macrophage migration inhibitory factor (MIF), and others were detected. The role of these molecules mostly consists in regulating the differentiation of Th1 and 17 cells, enhancing or modulating the inflammatory response [7,12,13,14,15,16]. Genetic background is central in JIA pathogenesis; monozygotic twins have a higher concordance rate, and siblings of affected individuals have a higher relative risk of illness [17]. The literature confirmed that the major histocompatibility complex (MHC) class II alleles are among the most influential factors in the pathogenesis of JIA [16,18,19,20]. Growing emerging research highlighted that class II human leukocyte antigen (HLA) molecules enhance disease susceptibility, indicating that the adaptive immunity also plays a crucial role in this class of diseases, and it is corroborated by the high rate of antinuclear antibodies (ANAs) in JIA patients [18,19,21]. A recent genome-wide association study (GWAS) has described novel non-HLA genes associated with JIA and emphasised the involvement of Th17 cell differentiation in the development of this group of disorders [9]. The characterisation of proinflammatory cytokines and chemokine profiles could provide a better understanding of JIA. However, despite significant improvements over the last decade, JIA aetiology and molecular mechanisms remain not fully understood. The aim of this review is to offer an overview of the most important and up to date evidence about the pathogenesis of JIA. 

## 2. Interleukin Signature and Immune Cell Involvement

In JIA, several types of cells, such as macrophages, neutrophils, natural killer (NK) cells, and lymphocytes, secrete proinflammatory mediators, including interleukins and chemotactic cytokines, that attract and promote the differentiation of immune cells in the inflammation site, that in JIA is usually, but not only, represented by joints [22,23]. In children with a susceptible genetic background and predisposing environmental factors, both innate and adaptative immunity cells cooperate in causing the inflammatory process. This, in combination with an abnormal synoviocyte growth, leads to a thickening of the synovial membrane [7]. Furthermore, because of an increased production of proangiogenic molecules, such as osteopontin and vascular endothelial growth factor, the hypertrophic synovium membrane becomes hypervascularised and joint Doppler ultrasonography is used for the detection of this phenomenon [7]. This process causes the creation of an aberrant hyperplastic synovium layer, called pannus [24]. The hyperproduction of a variety of proinflammatory molecules, such as IL-1 and TNFα, which stimulates osteoclasts and the expression of local metalloproteinases (MMPs), results in significant cartilage stress and bone erosions, which can lead to irreversible damage and growth delay (Figure 2) [7,24,25]. Distinct cell involvement has been noted in various JIA subtypes. CD4+ Th cells play a key role in oJIA and pJIA, and the two diseases are known to share a molecular pathophysiology and common genetic background [19]. Human effector CD4+ Th cells may be divided into three subgroups, based on the immunological role, cytokines’ profile, and on transcription factor expression [26]. Th1 cells mostly protect the body from intracellular infections by primarily producing interferon (IFN)γ, showing typical cell markers, such as Tbet [27,28]. Th2 cells are crucial for helminth protection since they express many cytokines, such as IL-4, IL-5, IL-9, and IL-13 [29]. 

The last group consists of Th17 cells, which have a leading role in the development of various autoimmune and inflammatory disorders, mostly producing IL-17 and IL-22 [27,30]. However, despite their central involvement in the pathophysiology of JIA, Th17 are very rare at inflammatory sites if compared to Th1 cells, probably due to their self-regulatory mechanisms [27,31]. An important molecular plasticity has been recently attributed to Th17 cells [27]. It is known that Th17 cells develop the capacity to generate IFNγ when stimulated by local inflammatory cytokines, such as IL-12 and TNFα. In oJIA, Th cells have an intermediate phenotype, known as Th17/Th1, and generate both IFNγ and IL-17, but they may quickly stop secreting IL-17 and only continue to produce IFNγ [26,27,28,32]. These non-classic Th1 cells are derived from Th17 and, differently from the classic counterpart, they continue to express both Tbet, a classic Th1 cell marker, and typical Th17 cell markers, such as ROR-γT, CD161, and CCR6 [28,32,33]. Both non-classic Th1 cells and Th17/Th1 lymphocytes are highly present in the synovial fluid of oJIA children [27,32,33]. These mechanisms evidence the pathogenetic role of CD4+ Th cells in JIA pathophysiology and allow a better comprehension of the involvement of Th17 cells and their own ability to produce IFNγ, shifting toward the Th17/Th1 and the non-classic Th1 phenotype after appropriate cytokines’ stimulation (Figure 3) [27,32]. In contrast, Treg cells are inhibited and decreased in number, contributing to an uncontrolled joint inflammation. This disrupted immune balance results in an increased production of proinflammatory cytokines, such as IL-17, and the subsequent joint damage due to expression of MMPs and IL-6 by synoviocytes [34]. However, both innate and adaptative immunity appear to have a pivotal role in oJIA, indeed a high rate of ANA is observed in such patients and altered activated neutrophils and macrophages with impaired phagocytic capacities have recently been noted in inflamed synovial fluid [10,11]. In the meanwhile, accumulating data show that sJIA should be categorised as an autoinflammatory disorder owing to its clinical processes and genetic background. Uncontrolled activation of monocytes, macrophages, and neutrophils, followed by the upregulation of several proinflammatory cytokines, such as IL-1, IL-18, and IL-6, play a major role in this form of autoimmune disease [7,35]. Based on these findings, several interleukins have largely been studied due to their central involvement in JIA, and the most important are described below.

### 2.1. TNFα

TNFα is a major regulator of inflammation and lead actor in the pathogenesis of both inflammatory and autoimmune disorders [36]. TNFα signalling pathways are activated by its binding with two unique receptors (TNFR1 and TNFR2), resulting in a wide range of biological responses including cell differentiation, and proliferation. Its role has not been fully understood yet, but its contribution in the progression of the disease is well known [36]. TNFα is mostly released by macrophages and Th1 cells, and when excessively produced, it results in an excessive synthesis of cathepsins and MMPs by synovial fibroblasts, causing fibrosis, stricture formation, and joint erosion [36]. In psoriatic arthritis (PsA), activated dendritic cells and macrophages generate large amounts of TNFα and IL-23, which promote naïve T lymphocyte differentiation in Th17 cells, which subsequently secrete IL-17. TNFα and IL-17 also activate keratinocytes and stimulate osteoclasts, which increase synovial hyperplasia and angiogenesis [36,37,38]. Under the TNFα stimulation, Th1/Th17 cells also migrate and infiltrate the uvea, causing damage to the blood–retinal barrier, enhancing the retinal vasculature, and recruiting in this district other leukocytes such as monocytes, lymphocytes, and neutrophils, that cause chronic inflammation in non-infectious uveitis [39,40,41,42]. Genetic investigation in JIA patients confirms the role of TNFα in JIA pathophysiology. It has been observed that Caucasian patients carrying the TNFα-308GA/AA and TNFα-238GA genotypes are associated with a worse prognosis and a lower response to anti-TNFα drugs [43]. Noteworthily, TNFα inhibitors are the main biological drugs used in JIA, interfering both with the shift of Th17 to Th1 cells and the TNFα upregulation of the synovial fibroblasts [27]. Their proven efficiency in improving the disease outcome of JIA patients underlines the key role of this cytokine [44,45,46,47]. 

### 2.2. IL-1 

Tissue macrophages, blood monocytes, and dendritic cells represent the major sources of IL-1β [48]. IL-1β plays a key role in the synthesis of powerful inflammatory mediators such as cyclooxygenase type 2, phospholipase A, and inducible nitric oxide (NO) synthase, which accounts for prostaglandin-E2, platelet activating factor, and NO production, in addition to angiogenic properties [48]. The activation of IL-1β consists in its synthesis and release, the activation of membrane receptor binding, and intracellular signal transduction, that leads to a complex sequence of events such as phosphorylation and ubiquitination, and results in the activation of nuclear factor kappa B (NF-kB), and the subsequent expression of proinflammatory cytokines, chemokines, and secondary mediators of inflammatory response in sJIA [49]. The key role of IL-1β in sJIA is well known. In peripheral blood, mononuclear cells from sJIA patients produce excessive levels of IL-1β compared with healthy controls [50]. Based on further understanding of sJIA pathogenesis, biological therapies target specific cytokines. Of these biologics, the IL-1 inhibitor canakinumab is currently approved by both the US Food and Drug Administration (FDA) and European Medicines Agency (EMA) for sJIA in children ≥ 2 years. Currently, the short-life IL-1 inhibitor anakinra, a recombinant IL-1 receptor antagonist, is approved by the EMA in sJIA patients aged 8 months or older and with a weight of 10 kg or above.

### 2.3. IL-6, IL-17, and IL-23

IL-6 is a pleiotropic cytokine that takes an active role in the acute phase of the pathology and has been detected at high concentrations both in blood and synovial fluid of JIA patients [51,52]. Its role includes osteoclast activation and joint inflammation, amplifying cytokine expression by peripheral monocytes [53]. It correlates with CRP, iron, haemoglobin, and platelet levels and with the number of affected joints and the degree of disability [54]. IL-6 plays a crucial role in oJIA and pJIA, probably because in such forms it has been shown to help to differentiate Th17 cells. Furthermore, IL-6 inhibitors have been found to improve the modulatory function of Treg cells in oJIA [55]. The key involvement of IL-6 in sJIA is supported by the fact that treatment with IL-6 inhibitors restores NK cytotoxic activity in affected patients [56]. Conversely, IL-6-expressing cells are poorly detected within entheses or subchondral bone lesions in spondyloarthritis, such as ankylosing spondylitis [57]. These findings confirm that the role of IL-6 in PsA and ErA is totally distinct from that in oJIA and pJIA, and this could be the likely cause of the poor efficacy of IL-6 inhibitors in such forms. Otherwise, ErA seems to be associated with HLA-B27-dependent T cell activation and the subsequent production of IL-17 and IL-23 [58]. IL-23 is produced by activated antigen-presenting cells (APCs), playing an important role in Th17 cell development and maintenance, and affects the pathogenicity of Th17 cells interacting with IL-17 and TNFα. IL-23 can induce the differentiation of Th17 cells, increasing IL-17 levels (IL-23/IL-17 axis) and leading to osteoclastogenesis [59,60]. With regard to IL-17, it promotes tissue inflammation. The IL-17A receptor is expressed in several cell types, including fibroblasts, osteoclasts, osteoblasts, monocytes, and synoviocytes [61]. IL-17A may induce the production of various chemokines, such as CXCL1, 5, and 10 and CCL2 and 20, also stimulating the secretion of other proinflammatory cytokines such as IL-6, TNF, and IL-1 [62,63,64]. In recent years, it has emerged that members of the IL-17 family are crucially involved in the pathogenesis of some subtypes of JIA, confirming that both oJIA and pJIA patients have a large amount of this cytokine in the active phase of the disease [14]. These data were recently confirmed in oJIA and rheumatoid factor (RF)-negative pJIA, leading to speculation that these two subtypes of JIA may represent a continuum of the same disease [8,65]. In another study, IL-17 levels were increased in the synovial fluid of patients with ERA and pJIA, as compared to subjects with sJIA [66]. Based on the studies showing the role of IL-17 in the pathogenesis of JIA and the role of the “IL-23/IL-17” axis, clinical studies were initially conducted to evaluate the effect of IL-23 inhibitors in the treatment of this disease [67].

### 2.4. IL-10

IL-10 is a pleomorphic cytokine produced by a wide variety of both innate and adaptive immune cells, including granulocytes, dendritic cells, macrophages, and T and B cells [68,69]. IL-10 modulates the local cytokine microenvironment and limits antigen presentation, preventing the efficient development of T cell responses and limiting basic microbicidal mechanisms [70]. It has been implicated in several major diseases such as systemic lupus erythematosus (SLE), rheumatoid arthritis (RA), and systemic sclerosis (SSc). Recent evidence has established that IL-10 is increased in blood of adult-onset Still’s disease (AOSD) patients and positively correlates with disease activity [71]. sJIA shares many clinical similarities with AOSD, despite the younger age onset, indeed, both are multifactorial autoinflammatory diseases with inappropriate activation of the inflammatory cascade and inflammatory cytokine production [72,73]. IL-10 could be a key factor in sJIA pathogenesis. This cytokine has been noted to be increased in sJIA patients compared to controls [70]. Furthermore, serum IL-10 levels are higher during the active phase compared to inactive phase of the disease [70]. These findings strongly support the theory that IL-10 has an important role in sJIA pathogenesis and could be a good marker for monitoring disease activity in sJIA, and it could probably also have a prognostic role. In addition, novel evidence has shown that IL-10 levels at diagnosis are significantly higher in sJIA patients with a polycyclic or persistent disease course compared to patients affected by the monocyclic form [70].

### 2.5. IL-21

T follicular helper (Tfh) cells are a novel subgroup of CD4+ Th cells, secreting IL-21 as a signature cytokine [74,75]. The role of the subset of Tfh cells has been confirmed in several autoimmune diseases and they provide help to antibody-producing B cells by provision of IL-10 and particularly IL-21 [75]. IL-21-secreting CD4+ Tfh cells are particularly enriched in joints of ANA + oJIA and pJIA patients, and low in ErA patients, in which Th17 cells have a major role instead [74]. However, further investigations are necessary to better understand the role of this cytokine and future therapeutic perspectives.

## 3. Genetic Background

The pathophysiology of JIA is characterised by a strong genetic background [76]. Epidemiological studies have reported significant concordance rates in monozygotic twins of approximately 25-40%, with a much greater risk than in other children [77]. First degree relatives of JIA patients show an increased prevalence of autoimmunity disorders, and the frequency of JIA among siblings is around 15–30 times that of the general population, with a wide similarity in disease onset and course [78,79,80]. Both HLA and non-HLA genes play their own role in susceptibility to JIA. Although a huge number of non-HLA candidate genes had been investigated in the past for associations with JIA, only a restricted number of associations were identified. Recently, GWASs have led to a better understanding of many novel non-HLA loci associated with JIA [81,82,83,84]. Moreover, many environmental factors have been widely studied, with unclear conclusions.

### 3.1. HLA Haplotypes and JIA Subtypes, an Overview

For several years, the MHC region has been considered as the most significant genetic risk factor for JIA. The development of innovative approaches for the analysis of conventional HLA alleles from genotyping array data is providing more detailed knowledge of the JIA molecular background [85]. The important HLA contribution to JIA development is estimated at a rate of around 20% [76]. The largest study evaluating the connection between HLA haplotypes and JIA subtypes evidenced both differences and similarities across the various forms of the disease [85]. The large HLA complex includes genes encoding class I (HLA A, B, and C), and class II (HLA-DR, DP, and DQ) molecules [76]. HLA-DR is an MHC class II cell surface receptor expressed by APCs such as lymphocytes and macrophages, that plays a very key role in the immune system’s transmission of peptides derived from extracellular proteins to Th lymphocytes. It is known that the greatest genetic relationship exists between MHC and adult RA risk and the presence of HLA-DRB1 has been linked to RA development, and particularly its amino acid sequence changes at positions 11 and 13 [86]. In JIA patients, the most clinical and immunological homogeneity has been noted between oJIA and RF − pJIA forms. These categories appear genetically quite similar, sharing a strong association for the haplotype HLA-DRB1*08 [19,85,87]. The amino acid position 13 is a key locus in the pathophysiology of these forms, and specifically the presence of a glycine in this position is associated with a greater risk of both oJIA and RF − pJIA, while histidine seems to be the most important risk factor for RF + pJIA [19,85,87]. Interestingly, the relationship between HLA-DRB1 and amino acid position 13 was found significant for both JIA and adult-onset RA, even if this specific HLA genotype appears most influential in the development of the paediatric disease [85]. In addition, the link between JIA and amino acid variants seems to be more significant than that with the traditional HLA alleles [85]. However, oJIA has also been associated with HLA-DRB1*01, HLA-DRB1*11, HLA-DRB1*13, HLA-DPB1*02, and HLA-DQB1*04; while HLA-DRB1*04 and HLA-DRB1*07 appear to play a protecting role [19,76,88]. In addition, RF – pJIA has been also associated with HLA-DPB1*03, while RF + pJIA, which can be regarded as the paediatric counterpart of adult RA, has been associated with HLA-DRB1*01, HLA-DRB1*04, HLA-DQA1*03, and HLA-DQB1*03 [19,76,88]. Different HLA connections have been associated with other JIA subsets: in ErA, HLA-B*27 is the main haplotype involved, like in ankylosing spondylitis, probably an adult-onset form different from the paediatric juvenile spondyloarthropathy category [6,85]. No statistical correlation was detected for PsA, even if the main HLA alleles involved are HLA-DQA1*0401, HLA-DRB1*08, HLA-DQB1*0402, and, less frequently, HLA-B*27 [19,85]. Several studies have confirmed relevant associations for sJIA, and HLA-DRB1*11 seems the most significant haplotype involved in this specific subtype, even if associations with HLA-DRB1*04 and HLA-DQA1*05 have also been reported [76,85,89]. However, this relation appears quite different to the other forms, confirming that sJIA has strong autoinflammatory characteristics and probably a different underlying genetic background. The assessment of the HLA expression confirms the clinical homogeneity between some categories, such as oJIA and RF − pJIA, and the heterogeneous aetiology of other different JIA subtypes, such as ErA and PsA. Even in the evaluation of HLA haplotypes, RF + pJIA shares some characteristics in common with adult RA. These findings help to better understand the pathogenesis of JIA and suggest that the current ILAR classification is not accurate enough and new classification criteria, giving more relevance to pathophysiological mechanisms and evidence-based improvements, are needed [85].

### 3.2. A Genome-Wide Study Approach for Non-HLA Genes Related to JIA

Two associated non-HLA loci were found in JIA genetic investigations prior to 2013: *PTPN22* and *PTPN2* [90]. Protein tyrosine phosphatase non-receptor type 22 (*PTPN22*) is a gene located on chromosome 1p13.2, encoding the homonym protein PTPN22, also called “lymphoid phosphatase” (LYP) in humans, which is expressed nearly exclusively by haematopoietic cells, especially T lymphocytes, in the thymus and spleen [91]. LYP is regarded as an inhibitor molecule of T cells, probably through its P1 motif that directly binds the Csk phosphorylates, a stunning negative regulator of T cell receptor (TCR)-mediated signalling, even if the exact mechanism of the Csk-PTPN22 complex is still not totally clear [91]. *PTPN22*-knockout mice exhibit increased T cell dependent immunoglobulin response to antigens, as well as enhanced effector and memory T cell proliferation [91,92,93]. Moreover, PTPN22-deficient Treg cells show increased adhesion properties [92]. A *PTPN22* missense mutation R620W (c.1858C > T, p.Arg620Trp) has been associated with an increased risk of several autoimmune disorders, such as SLE, type 1 diabetes mellitus (T1DM), vitiligo, RA, and finally JIA, probably by not promoting type 1 IFN-driven inhibition of inflammation [91,94,95,96,97,98,99]. R620W substitution affects the P1 motif, causing an impaired interaction with Csk, and therefore a reduced inhibition of T cell regulatory pathways [91]. However, R620W should be better considered a single nucleotide polymorphism (SNP), with an estimated prevalence ranging up to >10% in Northern Europe, around 7–8% in Western Europe, and <1% in Africa, Middle East Europe, America, and Asia [91,100]. *PTPN2* is a gene located on chromosome 18p11.21, encoding the homonym tyrosine phosphatase PTPN2, which also has its own role in the regulation of both T and B lymphocytes like PTPN22, modulating immunity through the Janus kinases/signal transducer and activator of transcription proteins (JAK/STAT) signalling pathways [101]. Haematopoietic tissues exhibit high levels of PTPN2, that seems to have an impact on a wide group of cells involved in immune system development. PTPN2-deficient mice exhibit increased levels of TNFα, a central proinflammatory cytokine in JIA [101]. In recent years, genome-wide association studies have defined a wide number of JIA-linked non-HLA loci [9,83,84,87,90,102]. Nevertheless, most of these relationships need to be validated by wider, more precise experimental research, even if convincing causal connection may be plausible when the tagging SNP corresponds with a missense coding variation [90,103]. The main genes investigated are those with a potential pathogenetic relevance, due to the presence of coding variations for which the causal gene can be determined with reasonable confidence. These data have been assessed by using a large variety of different genetic techniques, including the unbiased approach of GWAS and candidate gene approaches. The most important non-HLA genes that have been associated with JIA susceptibility are listed in Appendix A [81,82,83,84,87,90,102,104,105]. Many of these genes are widely expressed in haematopoietic tissues, including neutrophils and monocytes, and generally involved in T cell regulation and modulation, underlying once again the central role of both innate immune responses and adaptive immunity, through the modulation of many proinflammatory cytokines, including TNFα, highlighting interactions with each other in JIA pathophysiology [90]. Interestingly, oJIA and RF – pJIA share a common group of susceptibility loci, suggesting once again that these two subtypes could rather be the same disease, and that the number of affected joints should not even be considered a classification criterion, as suggested by Martini et al. [3]. Indeed, susceptibility to oJIA and RF − pJIA has been reported in several common non-HLA loci, including *PTPN22*, *PTPN2*, *STAT4*, *C12orf30*, *COG6*, *ANGPT1*, *ADAD1*-*IL2*-*IL21* [84,101]. According to the new JIA classification criteria in progress, the diagnosis of RF + pJIA will be possible even with positive anticitrullinated peptide antibodies (ACPAs) [3]. This consideration can be helpful to understand how much this disorder is considered as the paediatric counterpart of adult RA. These two subtypes share a quite similar genetic background in addition to clinical features, and many susceptibility loci are in common, including *TNFAIP3*, *PTPN22*, and *STAT4* [84,106]. In addition, no single genes have been evaluated for pJIA alone in the various studies, likely due to the pathophysiological differences between the RF− and RF+ polyarticular forms, which represent two distinct diseases, as outlined in the new JIA classification criteria that are currently being developed [3]. Many genes have been evaluated for sJIA susceptibility, but none of them has been found in common with other different JIA subtypes, evidencing the different genetic background of this form from the others (Appendix A). Nevertheless, prevalence of JIA is globally low, and sample sizes for GWAS analysis are limited, forcing studies to be conducted on cohorts of mixed JIA subgroups [84]. However, the purpose of this review is not to describe all of these genes in detail, but rather to focus in more detail on historical genes associated with JIA susceptibility, such as *PTPN22* and *PTPN2*, whose molecular aspects are better understood, and on genes related not to simple susceptibility but to monogenic forms of the disease.

### 3.3. Monogenic Forms of JIA

In view of the uncertainty surrounding JIA’s pathophysiology and categorisation, the discovery of four monogenic variants of JIA attributed to the genes *LACC1*, *LRBA*, *NFIL3*, and *UNC13D* has significantly increased the understanding of crucial molecular pathways of JIA [84]. The laccase domain-containing 1 (*LACC1*) gene encodes C13orf31/FAMIN, a key protein in the modulation of mitochondrial functionality [107]. Several different subtypes of JIA, including oJIA, RF – pJIA, sJIA, and ErA, have been described in association with autosomal recessive (AR) inherited SNP occurring in *LACC1* [108,109,110]. The exact role of FAMIN is not fully understood, but it forms a dimeric complex with fatty acid synthase (FASN) on peroxisome, promoting both lipogenesis and ATP regeneration [84,107]. The pathogenic *LACC1* mutation I254V (c.760A > G, p.Ile254Val) causes a serious impairment of FAMIN and is considered associated with the development of non-systemic JIA, inflammatory bowel diseases, and Behçet disease [84,111]. In addition, other important *LACC1* pathogenic mutations have been associated with JIA, such as C284R (c.850T > C, p.Cys284Arg), which has been found in several consanguineous sJIA patients, and the frameshift p.Cys43Tyrfs*6 and p.T276fs*2 mutations, which have been confirmed respectively in severe oJIA and RF − pJIA patients [84,109]. A second monogenic form of JIA is that related to AR mutations occurring in the lipopolysaccharide (LPS)-responsive beige-like anchor (*LRBA*) gene. It encodes the protein LRBA, a central molecule in the negative modulation of the degradation of cytotoxic T-lymphocyte-associated antigen 4 (CTLA-4) proteins [84]. CTLA-4 reduces the activation of T lymphocytes by competing with their own protein cluster of differentiation 28 (CD28), necessary for the correct functioning of T cells through the TCR costimulation [84,112,113]. Growing evidence is proving that *LRBA* could be a lead actor in the autoantigen tolerance and immune system modulation. Its loss of function is responsible for a subset of common variable immunodeficiency (CVID), an immune dysregulation disorder in which up to 10% of patients develop an autoimmune disorder, and among these JIA is frequently observed, both in oligoarticular and polyarticular forms, probably due to a reduced CTLA-4 activity [112,114]. Recently, two monozygotic twin girls affected by oJIA have been discovered to be affected by a novel homozygous mutation M170I (c.510G>A, p.Met170Ile) in the gene *NFIL3*, which plays a role in the development and modulation of NK cells and cytokine production by Th2 cells [115]. Studies in vitro revealed a reduced stability of the homonym NFIL3 protein, and *NFIL3*-knockout mice showed increased IL-1β expression and more severe joint symptoms and disease course [84,115]. The unc-13 homolog D (*UNC13D*) gene is essential in the immunology synapse for the correct cytolytic granule secretion, and its reduced expression has been found to be associated with a decreased NK cell degranulation [84]. A heterozygous mutation occurring in a specific *UNC13D* intronic region has been regarded as responsible for development of sJIA and recurrent macrophage activation syndrome (MAS), probably due to an impaired lymphocyte-specific downregulation of the NF-kB [116,117,118]. However, despite the discovery of these four monogenic forms, JIA should be correctly considered a polygenic illness, due to complex interaction between a susceptible genetic background, involving both HLA and non-HLA alleles, and multiple environmental factors. 

### 3.4. Environmental Factors

Together with the genetic susceptibility, several environmental factors may trigger and influence JIA with different mechanisms, including epigenetic changes, imbalance of microbiota, and immunity modulation [119]. Breastmilk has been related to a protective effect on JIA development [119]. Even if studies are controversial, a beneficial effect could be provided by breastfeeding due to probiotics and immunological protective factors, such as IL-10 and defensins [120]. In addition, early breastfeeding could modify the JIA phenotype and reduce disease severity [121]. Infectious agents could contribute to triggering JIA or influencing its course, even if there is currently only poor evidence. Interestingly, in countries where gastroenteritis is most common, ErA seem to have a higher incidence [119]. Prolonged antibiotic exposure has been associated with a higher development of JIA from two observational studies [122,123]. Given that smoking is the most important environmental risk factor for adult RA, the relationship between second-hand smoke exposure and JIA incidence has been evaluated, but inconclusive results have been reported [119,124]. Short-term pollutant exposure and bad air quality have been linked to JIA onset, but further evidence is needed [125,126]. The relationships between dietary habits and JIA, and residence and JIA, have also been evaluated, with inconsistent results [119]. No associations were detected with JIA and birth weight, even if pre-term children could have a higher risk to develop JIA [127]. No associations were detected between vitamin D serum levels and JIA development, even if a higher sun exposure during pregnancy could be a protective factor, through good levels of active vitamin D circulation [128,129]. In addition, small or inconsistent studies have focused on emotional stress and pet exposure during infancy [127,130]. 

## 4. Conclusions

JIA is a complex paediatric disease with multiple subtypes. Understanding the differences in the genetic background of the various subtypes, evaluating both HLA haplotypes and non-HLA genes, is critical to providing a better understanding of these disorders. It is known that oJIA and RF − pJIA share several genetic similarities in susceptible HLA haplotypes (mainly HLA-DRB1*08) and are characterised by a pathogenetic centrality of classic and non-classic Th1 cells and TNFα, while ErA appears to have peculiar molecular mechanisms, mainly involving HLA-B27 and IL-17. Although RF + pJIA may be considered the paediatric counterpart of adult RA, with a quite similar genetic background (such as the susceptibility for HLA-DRB1 and many clinical features), several differences have been highlighted with the other JIA subtypes, including the poor involvement of RF and ACPA in JIA pathogenesis, the centrality of CD4+ Th17/Th1 cells in oJIA, the importance of HLA-B27 in ErA and the autoinflammatory characteristics of sJIA. Furthermore, growing evidence suggests an inflammatory phenotype for sJIA, considering both genetic and molecular implications, such as the driving role of IL-1 and the different therapeutic strategies used to treat this form. Further genetic investigations and more detailed studies focusing on the molecular pathways of JIA will improve our understanding of the disease and help achieve more accurate classification criteria. A more comprehensive understanding of the molecular pathogenesis of JIA is a key goal to deliver more accurate targeted therapies and improve disease course and outcome.

## Figures and Tables

**Figure 1 ijms-24-01846-f001:**
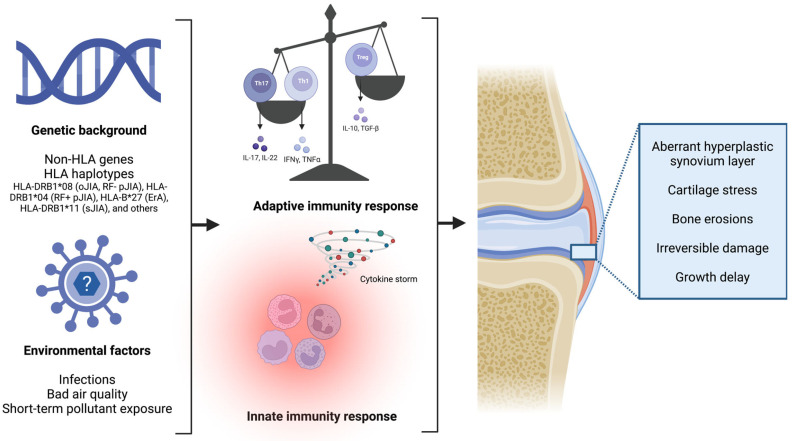
In children with a strong susceptible genetic background, a dysregulation of both the innate and adaptive immunity occurs, leading to cartilage stress and joint damage, up to permanent bone erosion damage (created with “BioRender.com”).

**Figure 2 ijms-24-01846-f002:**
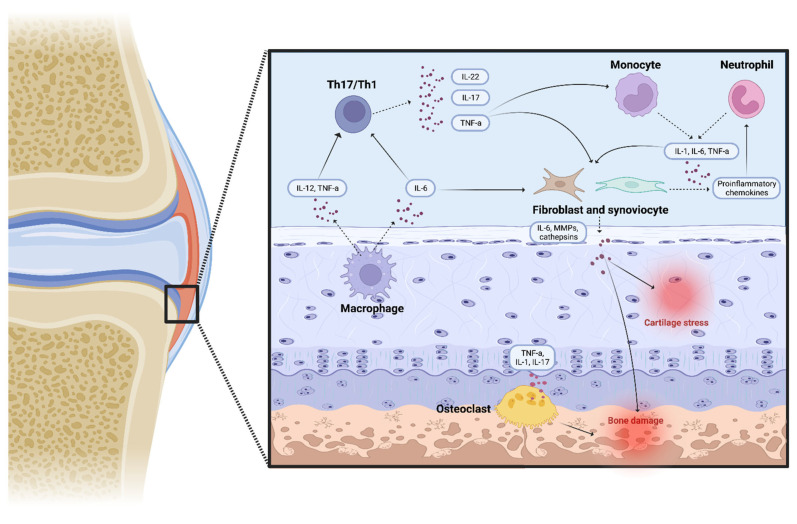
Several proinflammatory molecules and immune cells combine to form an inflamed environment in oJIA and pJIA, which leads to osteoclast activation, cartilage stress, and bone damage (created with “BioRender.com”).

**Figure 3 ijms-24-01846-f003:**
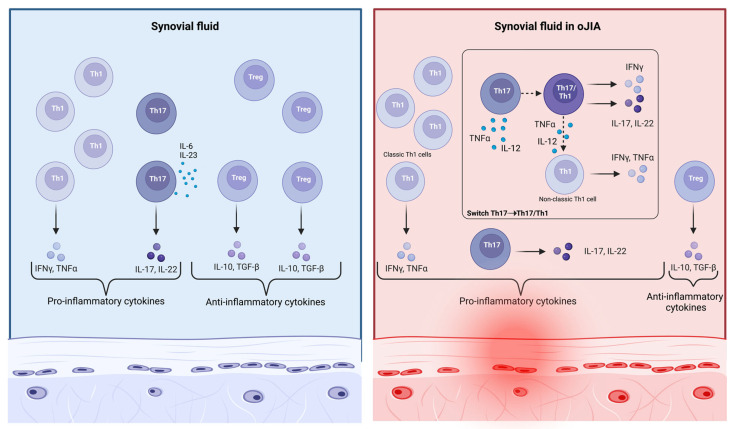
In the local inflammatory environment present in oJIA, CD4+ Th17 cells develop the ability to generate IFNγ when stimulated by cytokines such as IL-12 and TNFα. Such Th cells have an intermediate phenotype, known as Th17/Th1, and first generate both IFNγ and IL-17, and then only IFNγ. Both non-classic Th1 cells and Th17/Th1 cells are highly present in inflamed joints, with a large amount of IL-17, IFNγ, and TNFα (created with “BioRender.com”).

**Table 1 ijms-24-01846-t001:** Summary of arthritis characteristics and systemic features in the different JIA subtypes with a focus on cytokines involved in the aetiopathogenesis (adapted by Zaripova et al. [7]).

	sJIA	oJIA	RF − pJIA	RF + pJIA	ErA	PsA
Gender predominance	F = M	F	F	F	M	F = M
Arthritis presentation	Wrists, knees, and ankles are the joints commonly involved. Chronic pattern in 30–50% with slow development.	Number of joints affected: ≤4 Type of joints affected: large Asymmetric	Number of joints affected: ≥5 Type of joints affected: small and large	Number of joints affected: ≥5 Type of joints affected: small Erosive, aggressive symmetric polyarthritis	Lower limb joints are generally more affected Axial involvement	Type of joints affected: small and large Asymmetric
Systemic features	Fever, lymphadenopathy, evanescent rash, serositis, hepatosplenomegaly, MAS	30% uveitis	10% uveitis	10% uveitis Rheumatoid nodules	Acute anterior uveitis, enthesitis, gut inflammation	Psoriasis, dactylitis, onycholysis, nail pitting, uveitis (10–15%)
Adult counterpart	Adult-onset Still’s disease (AOSD)	-	RF − Rheumatoid arthritis	RF + Rheumatoid arthritis	Spondylo-arthropathies	Psoriatic arthritis
Type of disease	Autoinflammatory	Autoimmune	Autoimmune	Autoimmune	Autoimmune	Early onset—autoimmune, late onset—autoinflammatory
Biomarkers	Increased CRP, ferritin, platelets	70% ANA+	40% ANA+	RF+, ACPA+, ANA+ 40%	45–85% ANA+	50% ANA+
HLA genetic pre-disposition	HLA-DRB1*11 HLA-DBR1*04 HLA-DQA1*05	HLA-DRB1*08 HLA-DRB1*01 HLA-DRB1*11 HLA-DRB1*13, HLA-DPB1*02, HLA-DQB1*04	HLA DRB1*08 HLA-DPB1*03	HLA-DRB1*01 HLA-DRB1*04, HLA-DAQ1*03 HLA-DQB1*03 HLA-DRB1*08	HLA-B*27	HLA-DAQ1*0401 HLA-DRB1*08 HLA-DQB1*0402 Less frequently HLA-B*27
Immune system	Innate immune response	Adaptive immune system	Adaptive immune system	Adaptive immune system	Adaptive immune system	Early onset—adaptive immune system, late onset—innate immune response
Effector cells	Monocytes, macrophages, neutrophils	CD4+ and CD8+ T cells, neutrophils, T follicular helper	CD4+, CD8+ T cells, T follicular helper	CD4+ CD8+ T cells, T follicular helper	γδT cells, TH17 cells	TH1 and Th17 cell subsets, macrophages, and activated dendritic cells
Pathogenesis	Abnormal activation of phagocytes leads to hypersecretion of proinflammatory cytokines	Disrupted imbalance between inflammatory Th1/Th17 and Treg cells	Imbalance between inflammatory Th1/Th17 and Treg cells	Imbalance between inflammatory Th1/Th17 and Treg cells	HLA-B27 involved in presentation of arthritogenic peptide caused T cell activation	Autoinflammatory activation at the synovial–entheseal complex. Autoimmune processes in extra-articular tissues
Cytokines	IL-1, IL6, IL-18, IL-37, IL-10	TNFα, IL-17, IFNγ, IL-6, IL-21	TNFα, IL-17, IFNγ, IL-6, IL-33, IL-21	TNFα, IL-6, IL-17, IL-23, IL-21	TNFα, IL-17, IL-23	TNFα, IL-17, IL-23

Abbreviations. sJIA: systemic JIA, oJIA: oligoarticular JIA, RF: rheumatoid factor, ErA: enthesitis-related arthritis, pJIA: polyarticular JIA, PsA: psoriatic arthritis, MAS: macrophage activation syndrome, ANA: antinuclear antibodies, ACPA: anticitrullinated peptide antibodies, HLA: human leukocyte antigen, Th: T helper, TNF: tumour necrosis factor, IFN: interferon, IL: interleukin.

## Data Availability

Not applicable to this article as no datasets were generated or analysed during the current study.

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
