# Peer review of "Genetic Background and Molecular Mechanisms of Juvenile Idiopathic Arthritis"

_ijms, 2023, doi:10.3390/ijms24031846_

Round 1
Reviewer 1 Report
The authors describe exhaustively the molecular mechanism of JIA. The work is well developed and rich in detail. I propose only a few suggestions to make the work more all-encompassing:
1) In the "Environmental factors" section the authors should add possible influence of dietary factors, socioeconomic factors and child obesity.
2) In table 2 the authors should use "Single nucleotide polymorphisms" before abbreviation (SNP) as they did for "MAF"
3) In the figure 1, in genetic background, HLA haplotypes should be shortly expressed.
Author Response
Comment of the authors: Thank you for your precious suggestions and good comments. We have tried to correct the paper, as you have suggested.
Kind regards,
Dr. Saverio La Bella.
- In the "Environmental factors" section the authors should add possible influence of dietary factors, socioeconomic factors and child obesity.
Response 1: Thank you for your suggestion. Inconsistent results have been found between JIA development and food habits and residence. We have added this statement in the paragrapher. On the other hand, child obesity has been evaluated by several studied to its relation to disease activity and quality of life in JIA patients. However, no consistent results have been highlited and we found no studies evaluating the relation between juvenile obesity and susceptibility for JIA development.
- In table 2 the authors should use "Single nucleotide polymorphisms" before abbreviation (SNP) as they did for "MAF"
Response 2: Thank you for your comment. We have corrected this abbreviation and moved table 2 to a supplementary file, as the other reviewer has suggested.
- In the figure 1, in genetic background, HLA haplotypes should be shortly expressed.
Response 3: Thank you for your comment. We have updated a correct version of Figure 1, according to your suggestions and the comments of the other reviewer.
Reviewer 2 Report
This manuscript is a comprehensive review of the genetic background and molecular mechanisms of juvenile idiopathic arthritis. The text is well-organized and the description provides an excellent review of JIA. To further strengthen this article, here are major and minor comments.
Major
· Scope of the review: The abstract states that a deeper and broader understanding is critical, but it is also stated (lines 376-377) that ….. to focus in more detail on historical genes associated with JIA susceptibility. These two statements are not completely consistent. If the scope is not to review the potential role of many other genes, Table 2 is wasteful. This is a large table with a list of many genes, but no comprehensive analysis is conducted. The table can be compressed or moved to a supplementary table.
· Table 2: no single genes are listed for pJIA only, although there are many genes for oJIA only and sJIA only. Please describe whether this result was expected and if so why.
· Linkage to TNFa and cytokines: the genetic background for non-HLA genes, reviewed in this article, is not reviewed to link to the action of TNFa and cytokines. A pathway analysis of genes in Table 2, might be useful.
· Line 24: The last sentence in the abstract is “ …. to providing more accurate targeted therapies and improving disease outcome.” It is recommended to add the description to respond to this sentence.
Minor
· line 19: Please spell out “HLA” in the abstract.
· Table 1: “Effectors cells” should read “Effector cells.”
· Figure 1: In section 3.4, some of the environmental factors such as second-hand smoke exposure are not supported. It is fine to keep the current section 3.4, but it is recommended to remove those factors from Figure 1.
· Line 179: “represents” should read “represent.”
· Line 181: “type 2” is duplicated.
· Line 280: the reference [85] in 2017 may not be the recent study.
· Comparison to adult RA: The comparison can be expanded slightly to highlight JIA’s unique features, focusing on the genetic background and molecular mechanisms in conclusions.
Author Response
Comment of the authors: Thank you for your precious comments. Reading that our work has provided an excellent review of JIA has been an acknowledgment for our work.
Kind regards,
Dr. Saverio La Bella.
Major revisions:
- Scope of the review: The abstract states that a deeper and broader understanding is critical, but it is also stated (lines 376-377) that ….. to focus in more detail on historical genes associated with JIA susceptibility. These two statements are not completely consistent. If the scope is not to review the potential role of many other genes, Table 2 is wasteful. This is a large table with a list of many genes, but no comprehensive analysis is conducted. The table can be compressed or moved to a supplementary table.
And
Line 24: The last sentence in the abstract is “ …. to providing more accurate targeted therapies and improving disease outcome.” It is recommended to add the description to respond to this sentence.
Response 1: Thank you for your suggestion. The first statement has been removed and replaced with a different ending for the abstract. Table 2 has been moved to a supplementary file, as you suggested.
- Table 2: no single genes are listed for pJIA only, although there are many genes for oJIA only and sJIA only. Please describe whether this result was expected and if so why.
Response 2: Thank you for your comment. RF + pJIA and RF – pJIA represents two different diseases. Indeed, according to the new JIA classification criteria in progress, oJIA and RF – pJIA will be considered a new JIA subtype (early onset ANA + JIA) that will not evaluate the number of affected joints (https://pubmed.ncbi.nlm.nih.gov/30275259). RF + pJIA is the pediatric counterpart of the adult onset RA, as highlited, for example, by the common HLA haplotypes, and the clinical features (not discussed in this review). Indeed, ACPA will be soon considered for RF + JIA diagnosis in the new classification criteria, as in adult onset RA.
- Linkage to TNFa and cytokines: the genetic background for non-HLA genes, reviewed in this article, is not reviewed to link to the action of TNFa and cytokines. A pathway analysis of genes in Table 2, might be useful.
Response 3: Thank you for your suggestion. Many non-HLA genes related to JIA susceptibility are involved in the regulation of immune cells and expressed in hematopoietic tissues. Modulation of cytokines, including TNF alpha, is one of the molecular effects of all these genes. We added this statement in the text. The purpose of our article is to provide a narrative overview of the molecular pathogenesis of JIA, and Table 2 has been moved to a supplementary table, to make the text more descriptive and more readable in the article.
Minor revisions
- Line 19: Please spell out “HLA” in the abstract. - Table 1: “Effectors cells” should read “Effector cells.” - Figure 1: In section 3.4, some of the environmental factors such as second-hand smoke exposure are not supported. It is fine to keep the current section 3.4, but it is recommended to remove those factors from Figure 1 - Line 179: “represents” should read “represent.” - Line 181: “type 2” is duplicated - Line 280: the reference [85] in 2017 may not be the recent study.
Response 1: Thank you for your comment. These points have been corrected in the text, and Figure 1 has been updated as you and the other reviewer have suggested.
- Comparison to adult RA: The comparison can be expanded slightly to highlight JIA’s unique features, focusing on the genetic background and molecular mechanisms in conclusions.
Response 2: Thank you for your suggestion. The differences between RA and JIA pathogenesis have been summarized in the conclusions.